# The Microstructures and Mechanical Properties of a Welded Ni-Based Hastelloy X Superalloy

**Yuan Liu, Qingqing Ding \*, Xiao Wei, Yuefei Zhang, Ze Zhang \* and Hongbin Bei \***

School of Materials Science and Engineering, Zhejiang University, Hangzhou 310027, China
\* Correspondence: qq_ding@zju.edu.cn (Q.D.); zezhang@zju.edu.cn (Z.Z.); hbei2018@zju.edu.cn (H.B.)

**Abstract:** The Hastelloy X superalloy is a widely used solid-solution Ni-based sheet alloy for gas turbines, aero-engine combustion chambers, and other hot-end components. To investigate the effect of microstructure, especially grain size, on its weldability, Hastelloy X alloy bars are homogenized, cold-rolled to thin sheets, and recrystallized under different conditions to obtain equiaxed grain microstructures with average grain sizes of ~5 μm, ~12 μm, and ~90 μm. The laser welding process is used for joining the alloy sheets, and then the alloy's weldability is investigated through microstructural and mechanical property characterizations. The microstructures in weld consist of coarse columnar grains with dendrite, and grain sizes of these columnar grains are almost the same when grain size of Hastelloy X base metal increases from ~5 μm to ~90 μm. Moreover, although all welds exhibit lower yield strengths (YS), ultimate tensile strengths (UTS), and elongations to fracture (EF) than the base metal, the degrees of reduction in them become slight when the grain size of base metal increases from ~5 μm to ~90 μm.

**Keywords:** superalloy; laser welding; microstructure; grain size; mechanical properties

## 1. Introduction

The Hastelloy X superalloy is a nickel-chromium-iron-molybdenum alloy. As a typical Ni-based solid solution alloy, the Hastelloy X alloy possesses exceptional characteristics, including outstanding mechanical strength and ductility at high temperatures, good resistance to oxidation and corrosion, and excellent weldability [1–4]. As early as 1952, this alloy was first developed by Haynes International, INC for manufacturing combustor parts in jet aircraft. To date, it is still a widely used Ni-based solid solution alloy for gas turbines, aero-engine combustion chambers, and other hot-end components [5–7].

Various applications of the Hastelloy X alloy necessitate metallic components with large scales and complicated structures to satisfy diversified needs. Generally, large-scale metallic components are difficult for one-step shape-forming and are commonly joined by single parts made of specific alloys. Welding is a crucial fabrication technology for joining metallic parts into structures. Therefore, the weldability of an alloy must be investigated before it is adopted for practical application [8–11]. The Hastelloy X alloy is no exception. Previously, much research was conducted to determine the weldability of the alloy, showing its excellent welding characteristics [12–14]. However, previous research mainly focuses on investigating the effect of welding processes and welding parameters on its weldability, while the effect of microstructure, especially the grain size of the Hastelloy X alloy on its weldability is rarely mentioned.

As a solid-solution alloy, the microstructure of the Hastelloy X alloy, e.g., the grain size and elemental segregation, can significantly affect its mechanical properties [15–17]. During the welding process, the grain structure will change, and elemental segregation will take place in the fusion zone, and then affect mechanical properties of the welded alloy. Therefore, the microstructure and mechanical properties of the welded Hastelloy X alloy are worth investigation. Moreover, properties of welded alloys are also affected

by the welding process [18,19]. In previous studies, electron beam (EB) welding, seam welding, gas tungsten arc (GTA) welding, etc., were used to determine the weldability of the Hastelloy X alloy. Although its excellent welding characteristics make this alloy suitable for almost all the welding processes, these traditional welding processes have their own disadvantages, such as high heat input and low heat concentration, which might lead to cracks in the weld and even a severe distortion of the alloy. As a new and advanced welding process, laser welding has a number of appealing characteristics, including low heat input and high heat concentration. These unique advantages of laser welding make it a candidate for welding superalloys [20–26].

Here, we use the laser welding process to join the Hastelloy X alloy sheets, and then the microstructure and mechanical properties of the welded alloy are investigated systematically. Moreover, by analyzing the mechanical properties of the welded alloys, whose grain sizes of base metals are different, the effect of grain size on the alloy's weldability is comprehensively investigated in this study. Our study provides a representative example for the study of the microstructure and mechanical properties of the welded Hastelloy X superalloy, which might be significant for the application of laser welding to this alloy.

## 2. Materials and Methods

### 2.1. Material Preparation

The hot-rolled Hastelloy X alloy bars (purchased from Taizhou Boyan Metal Products CO., Ltd, Taizhou, China) are first homogenized at 1200 °C for 4 h, and then quenched in room-temperature (RT) water bath. Cold rolling is conducted for the homogenized alloys to reduce their thicknesses from ~8 mm to ~2 mm. The alloy sheets are recrystallized at 800 °C for 1 h, at 1100 °C for 0.5 h, and at 1200 °C for 1 h to obtain recrystallized sheets with different grain sizes. All heat treatments are conducted under an Ar atmosphere.

### 2.2. Laser Welding

An automatic laser welding machine (Hongyuan Layser, Suzhou, China) with maximum power of 1000 W is used for laser welding processes. Before welding, alloy sheets are ground to remove the oxide layer on the surface and then cleaned ultrasonically in ethanol. Table 1 shows the welding parameters used in the laser welding process. With such welding parameters, the laser beam spot diameter on the alloy sheet is $1.2 \pm 0.1$ mm.

**Table 1.** Parameters for the laser welding process in experiments.

| Parameters | Values |
|---|---|
| Constant Laser power (W) | 800 |
| Average Laser power (W) | 800 |
| Welding speed (mm/s) | 10 |
| Power density (W/mm$^2$) | 603 |
| Shielding gas type | Ar |

### 2.3. Tensile Testing

The recrystallized alloy sheets are cut into dog-bone-shape tensile specimens by electrical discharge machining (EDM). The gauge sections of tensile specimens are about 1.8 mm × 1.7 mm × 9.5 mm. The surfaces of gauge sections are ground by SiC papers of 400 and 600 grits to eliminate the oxide layer and damages caused by EDM. A screw-driven mechanical testing machine equipped with an induction heater is used to conduct tensile tests at different temperatures. For tensile tests at elevated temperatures, specimens are heated to the test temperature and retained for 15 min before adding stress. Tensile tests at each temperature are repeated more than three times to confirm the data reproducibility. For all tensile tests in this study, the crosshead displacement rate is fixed at 0.57 mm/min, corresponding to an engineering strain rate of $10^{-3} \cdot s^{-1}$.

### 2.4. Microstructure Characterization

To observe the microstructure, the Hastelloy X alloy samples are ground with SiC papers of 600, 800, and 1200 grits and then mechanically polished. Microstructural characterizations are conducted by using a HITACHI TM4000 plus tabletop microscope (Hitachi, Japan) and an FEI-quanta 650 FEG scanning electron microscope (FEI Company, Netherlands) equipped with a secondary electron detector, an Oxford energy dispersive spectroscopy (EDS) detector, and an Oxford electron backscattered diffraction (EBSD) detector. Back-scattered electron (BSE) micrographs and EBSD are used to characterize the grain structure; EDS mapping is used to examine the elemental segregation in the alloy.

### 3. Results

#### 3.1. The Initial Microstructures of Base Metal

Figure 1 is the representative microstructures and EDS maps of recrystallized Hastelloy X base metal sheets. Figure 1a–c show alloys' microstructure with annealing twins after recrystallization. The average grain sizes of base metals are determined by the linear intercept method, and they are found to be ~5 μm (Figure 1a), ~12 μm (Figure 1b), and ~90 μm (Figure 1c), corresponding to recrystallization at 800 °C/1 h, 1100 °C/0.5 h, and 1200 °C/1 h. Grain size increases with increasing recrystallization temperature and duration. Moreover, under the BSE imaging mode, particles with higher contrast are homogeneously distributed in alloys, especially in the alloys whose average grain sizes are ~5 μm (Figure 1a) and ~12 μm (Figure 1b). EDS results (Figure 1d) indicate that this phase with higher contrast is the Mo-rich phase.

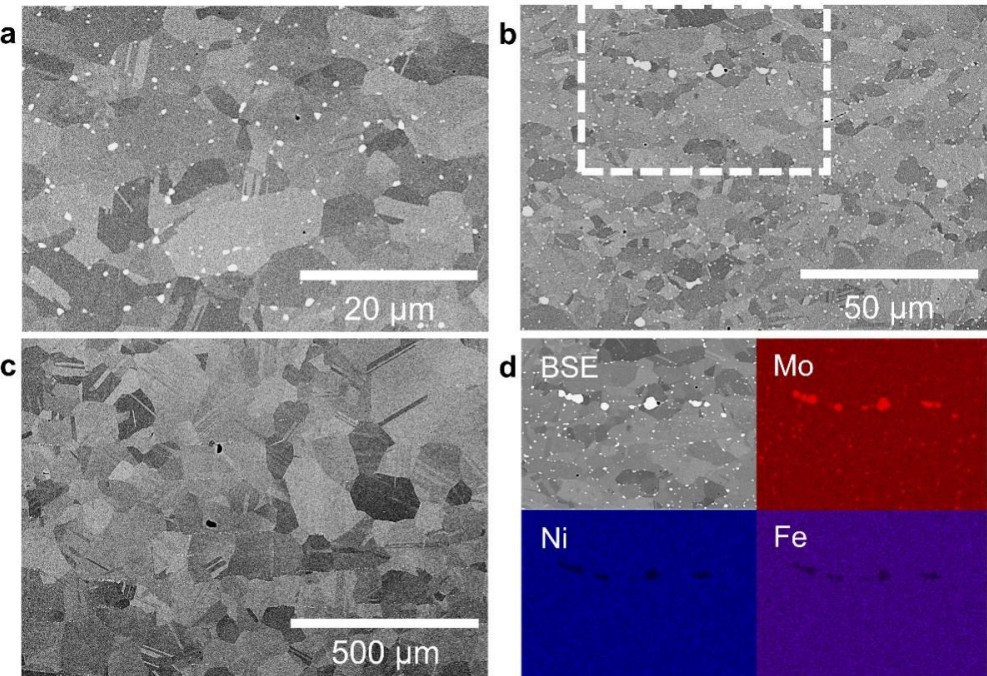

**Figure 1.** Grain structures and EDS maps of Hastelloy X base metals. (**a–c**) Equiaxed grain structures of the base metals after different heat treatments, with grain sizes of (**a**) ~5 μm, (**b**) ~12 μm, and (**c**) ~90 μm. (**d**) EDS maps of the area marked by white dash rectangle in (**b**) indicating that the phase with higher contrast is Mo-rich phase.

#### 3.2. The Tensile Properties of the Hastelloy X Base Metals

Tensile tests of Hastelloy X base metals are conducted to characterize mechanical properties of the alloy before laser welding. Figure 2a–c shows the engineering stress-strain curves of base metals with grain sizes of ~5 μm (Figure 2a), ~12 μm (Figure 2b), and ~90 μm (Figure 2c) at temperatures from 20 °C to 800 °C. Figure 2d–f summarize average values of

0.2% offset yield strength (YS), ultimate tensile strength (UTS), and elongation to fracture (EF) of base metals at various temperatures, with error bars included. In Figure 2d–f, strong temperature dependence trends can be observed for strength and elongation. For base metals with different grain sizes, the highest value of YS and UTS are all obtained at RT, and YS and UTS decrease when temperature increases from RT to 800 °C. Hastelloy X base metals show relatively high values of EF at RT (higher than 20%), and EF tends to increase with increasing temperature.

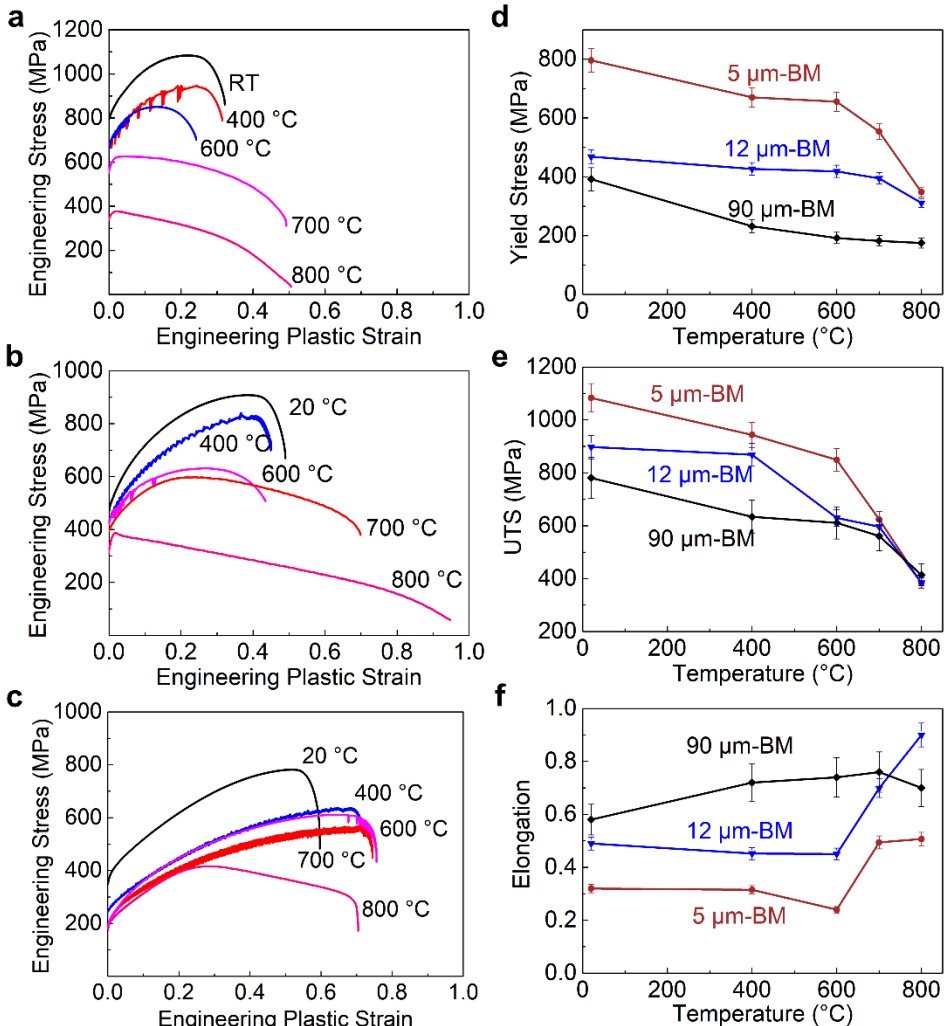

**Figure 2.** Tensile properties of the Hastelloy X base metals. Engineering stress-strain curves of the recrystallized Hastelloy X base metals with grain sizes of (**a**) ~5 μm; (**b**) ~12 μm; (**c**) ~90 μm at various temperatures from 20 °C to 800 °C. Average values of (**d**) YS; (**e**) UTS and (**f**) EF of Hastelloy X base metals at various temperatures from 20 °C to 800 °C.

It can also be found in Figure 2d–f that the grain size of Hastelloy X base metal significantly affects its strength and elongation. The YS and UTS of the base metal decrease simultaneously when the grain size increases from ~5 μm to ~90 μm at the same temperature (Figure 2d,e). Unlike YS and UTS, the EF of base metal increases when the grain size increases from ~5 μm to ~90 μm at the same temperature (Figure 2f). Therefore, a decrease in grain size of the base metal results in an increase in its strength, as well as a decrease in its elongation.

### 3.3. Microstructures of the Welded Hastelloy X Alloy

Figure 3a is the macrograph of a welded Hastelloy X alloy sheet welded by laser welding. BSE images of transverse sections of welds are shown in Figure 3b–d, corresponding to Hastelloy X base metals with grain sizes of ~5 μm (Figure 3b), ~12 μm (Figure 3c) and ~90 μm (Figure 3d). All the alloy sheets have been fully welded, and there is no cracks in the welds, indicating the exceptional weldability of the alloy. Although the grain sizes of base metals are different before welding, welds have microstructural similarities. For all welds shown in Figure 3b–d, fusion zones (FZ) exhibit grain structures of coarse columnar grains growing from fusion lines to centerlines. However, the grain structure at the center of the welds cannot be identified clearly from BSE micrographs in Figure 3b–d. Thus, the following microstructural characterizations are conducted to further investigate grain structures of the fusion zone for welded alloys.

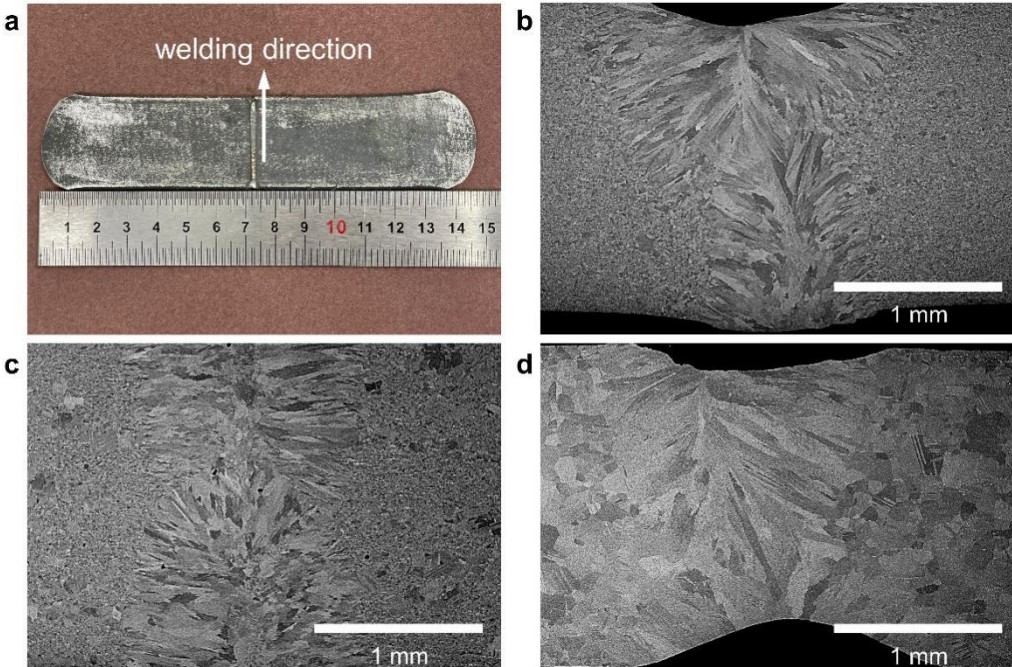

**Figure 3.** Photographs and microstructures of welded Hastelloy X alloys. (**a**) A photograph of a welded Hastelloy X alloy sheet. (**b–d**) BSE micrographs of welds' transverse sections of metals with grain sizes of (**b**) ~5 μm; (**c**) ~12 μm; (**d**) ~90 μm. All welds have no cracks and exhibit coarse columnar grains microstructure.

The weld of the Hastelloy X base metal with average grain sizes of ~12 μm is selected as a representative example for the weld of the small grain size base metal (Figure 4). From the EBSD figure (Figure 4b), three different types of microstructures in welded alloy can be clearly observed. The equiaxed grain not affected by laser welding can be observed outside the weld, indicated by a white arrow in Figure 4b. In the weld, columnar grains grow from the fusion lines to the centerline, indicated by a green arrow in Figure 4b. In the process of welding solidification, grains tend to grow along the direction of the maximum temperature gradient, toward the solid-liquid interface. Therefore, columnar grains in the weld grow from the fusion line towards the centerline, which results in the typical columnar grain structure of the welded Hastelloy X alloy.

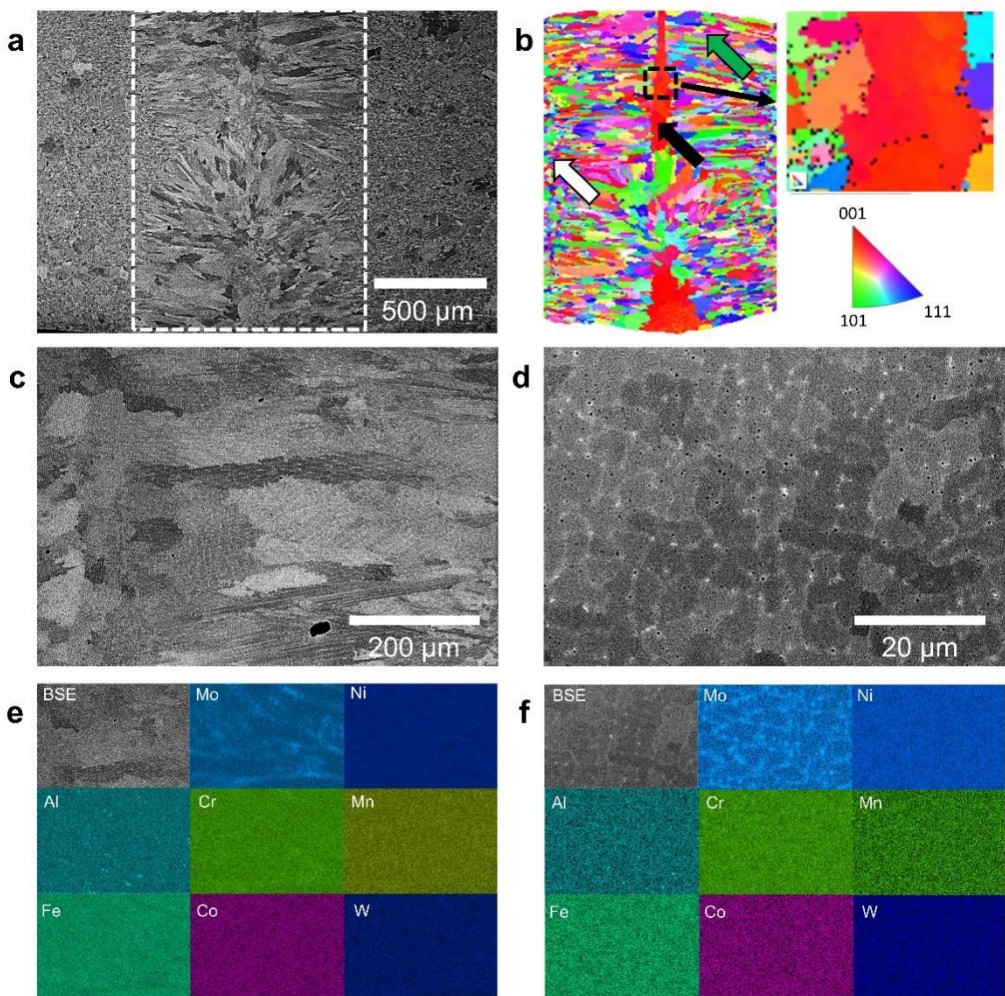

**Figure 4.** The BSE micrographs, EBSD, and EDS mappings of the welded Hastelloy X alloy when the grain size of base metal is ~12 μm. (**a**) Low-magnification cross-section image of the weld. (**b**) EBSD of the area marked by a white dash rectangle in (**a**). (**c**,**d**) BSE micrographs of columnar grains from the fusion lines to the centerline (**c**) and at the weld center (**d**). (**e**,**f**) EDS maps of columnar grains shown in (**c**,**d**), indicating that solidification segregations with higher contrast in the weld are all Mo-rich phase.

At the weld center, there is a coarse columnar grain growing from the surface of the alloy sheet towards the inside of FZ, indicated by a black arrow in Figure 4b, which might result from the heat flow from the center to the surface. Average lengths and aspect ratios of columnar grains in the welds are measured. It is found that the average lengths of columnar grains are ~250 μm, and the average aspect ratios (the ratio of the length and width of grains) are ~4 for welds whose grain sizes of base metals are ~5 μm and ~12 μm, while the average length of columnar grains is ~270 μm, and the average aspect ratio is ~4.3 for the weld whose grain size of base metal is ~90 μm. Therefore, the grain size of the base metal does not change the columnar grain structure significantly.

In addition, Figure 4c,d indicate that solidification segregations exist in columnar grains in the fusion zone. EDS maps (Figure 4e,f) reveal that these segregations are Mo-rich, which is similar to the Mo-rich phase in the as-cast Hastelloy X alloy [5].

The weld of the Hastelloy X base metal with an average grain size of ~90 μm is also characterized (Figure 5). Figure 5b shows the EBSD of the area marked with a white dash rectangle in Figure 5a. Similar to the welding microstructure shown in Figure 4b, the base metal outside the weld maintains an equiaxed grain microstructure (indicated by a white

arrow in Figure 5b); in the weld, columnar grains growing from the fusion lines to the centerline can be observed (indicated by a green arrow); at the weld center, the columnar grain growing from the surface of the alloy sheet towards the inside of FZ is observed (indicated by a black arrow). It indicates that although the grain sizes of base metals are different (e.g., ~12 μm and ~90 μm), the microstructures in the welds are similar. Similarly, solidification segregations are also observed in the fusion zone of the weld (Figure 5c,d). EDS maps (Figure 5e,f) again indicate that they are all Mo-rich phases.

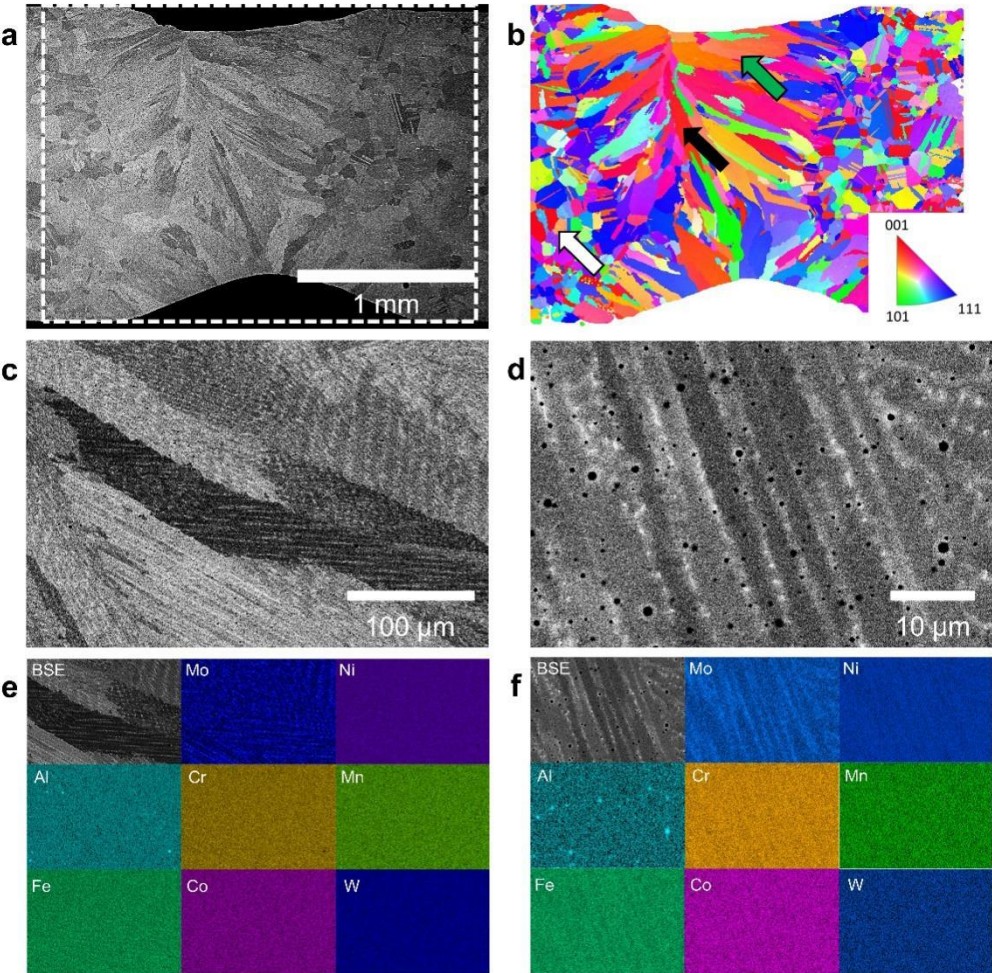

**Figure 5.** The BSE micrographs, EBSD, and EDS mappings of the welded Hastelloy X alloy when the grain size of the base metal is ~90 μm. (**a**) Low-magnification cross-section image of the weld. (**b**) EBSD of the area marked by a white dash rectangle in (**a**). (**c,d**) BSE micrographs of columnar grains from the fusion lines to the centerline (**c**) and at the weld center (**d**). (**e,f**) EDS mappings of columnar grains shown in (**c,d**), indicating that phases with higher contrast in the weld are all in the Mo-rich phase.

### 3.4. Mechanical Properties of Welded Hastelloy X Alloys

Figure 6a–c show the engineering stress-strain curves of welded Hastelloy X alloys when the average grain size of the base metal is ~5 μm (Figure 6a), ~12 μm (Figure 6b), and ~90 μm (Figure 6c), respectively. A strong temperature dependence trend can be observed for strength and elongation of the welded alloys. YS and UTS simultaneously decrease, while EF increases when the test temperature increases from 20 °C to 800 °C. The temperature dependence of the strength and elongation for the welded Hastelloy X alloy are similar to that of base metal.

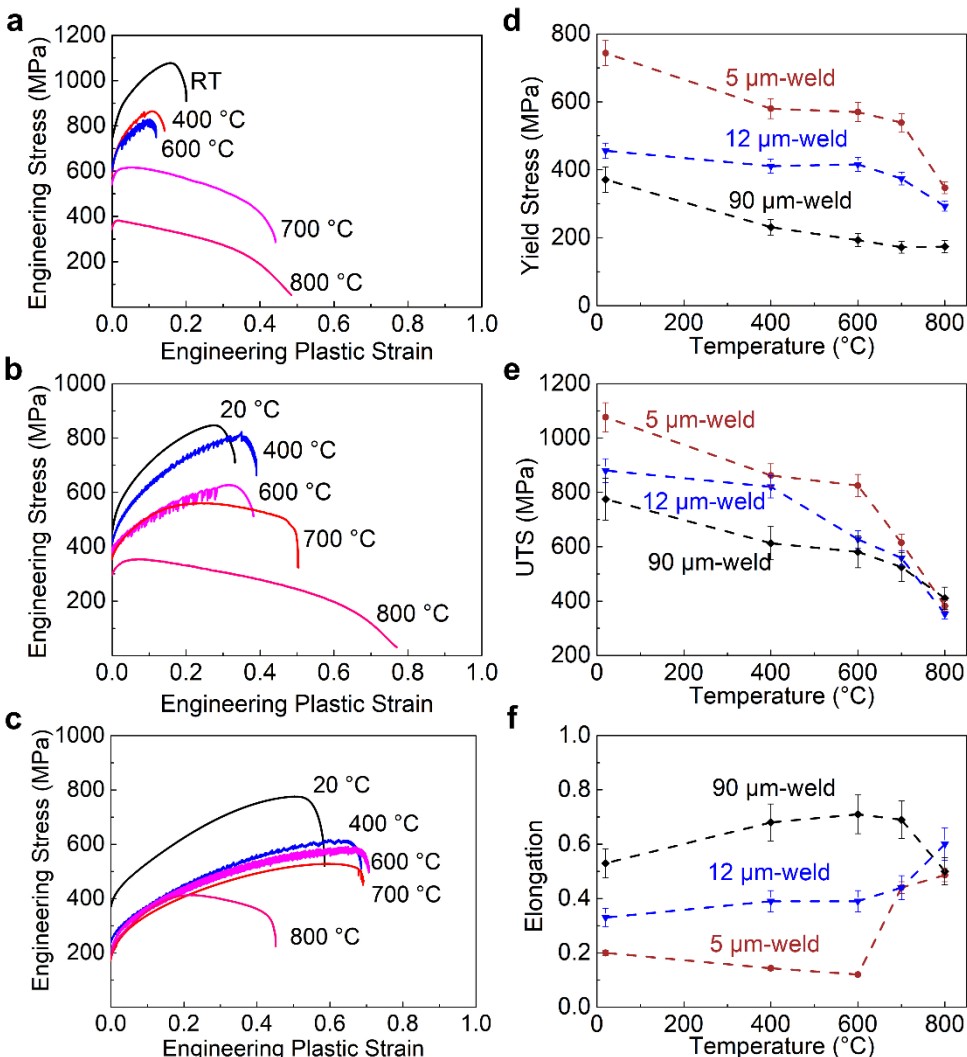

**Figure 6.** Mechanical properties of welded Hastelloy X alloys. Engineering stress-strain curves of welded alloys when the grain size of the base metal is (**a**) ~5 μm; (**b**) ~12 μm; (**c**) ~90 μm at various temperatures from 20 °C to 800 °C. Average values of (**d**) YS, (**e**) UTS, and (**f**) EF of welded Hastelloy X alloy at various temperatures from 20 °C to 800 °C.

Moreover, mechanical properties of welded Hastelloy X alloy are affected by the average grain size of the base metal. For instance, at RT, the YS of the welded alloy decreases by ~50%, and the UTS decreases by ~30% when the grain size of base metal increases from ~5 μm to ~90 μm (Figure 6d,e). The EF of the welded alloy increases by ~150% when the grain size of base metal increases from ~5 μm to ~90 μm (Figure 6f). The results demonstrate that the YS and UTS of the welded alloy decrease, while the EF increases, when the grain size of the base metal increases.

## 4. Discussion

### 4.1. Fracture Analysis

Microstructural characterization of fractures is performed to identify possible causes for the fracture of Hastelloy X base metals and the laser-welded alloys. For the welded tensile specimens, fracture occurs in the welds (Figure 7a). The microstructures near the fracture are typical columnar grains, indicated by the black arrow in Figure 7a. The fracture occurs within the columnar grain at the weld center.

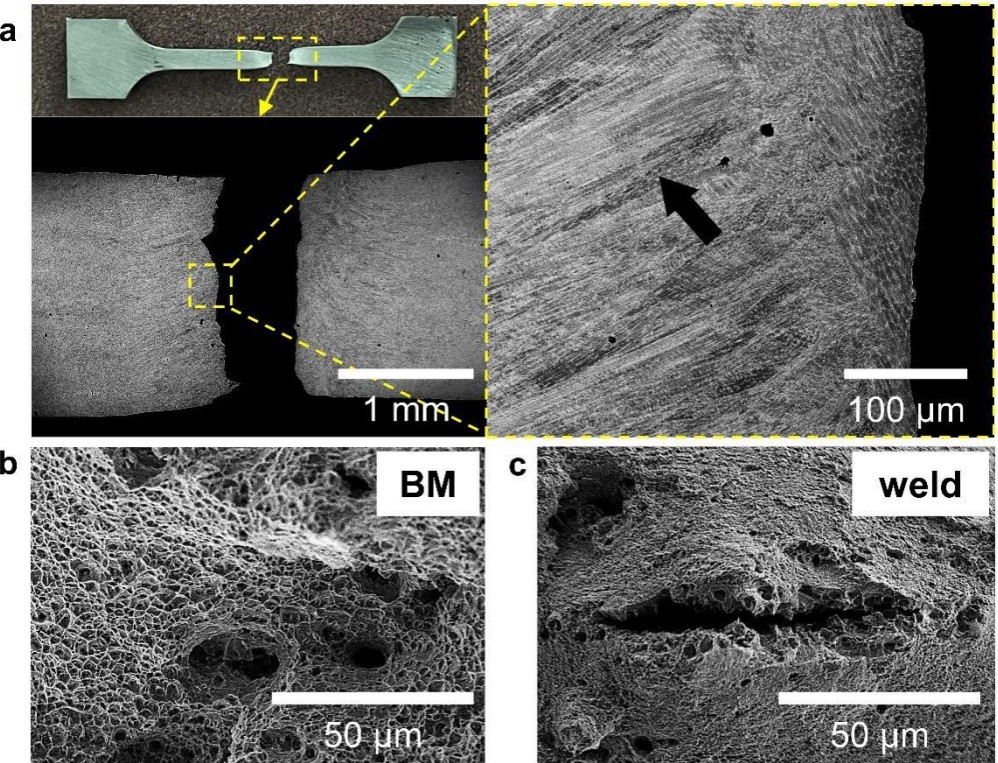

**Figure 7.** Microstructure of fractures of base metals and welded alloys. (**a**) Photograph and corresponding BSE image of the tensile fractured welded specimen whose grain size of base metal is ~12 μm. (**b**) SE images of fractures of base metal (**b**) whose grain size is ~12 μm and its corresponding welded specimen (**c**). Dimples on fractures demonstrate that fractures of both base metal and welded alloy are ductile.

Figure 7b is the secondary electron (SE) micrograph, showing RT fracture surface of Hastelloy X base metal. Dimples can be observed on the fracture surface, showing that the fracture of base metal is a ductile fracture at RT. For the fracture of welded Hastelloy X alloy (Figure 7c), although the EF is reduced, similar dimples can also be observed on the fracture surface, revealing that the fracture mode is also ductile.

### 4.2. Mechanical Behavior

As observed in Figures 2 and 6, the YS and UTS of base metal decrease, while EF increases when the grain size of Hastelloy X base metal increases from ~5 μm to ~90 μm. A similar case is also observed for the welded alloy. For better comparison, the average values of YS, UTS, and EF of Hastelloy X base metals and welded alloys with different grain sizes are summarized in Figure 8. It is found that YS, UTS, and EF all decrease after laser welding, and degrees of reduction are different when the grain sizes of the base metal change.

Quantitatively, at RT, the welded Hastelloy X alloy maintains ~93% of the YS of the base metal when the grain size of base metal is ~5 μm; when the grain size of base metal is ~12 μm, the laser welded alloy maintains ~96% of its YS; when the grain size of base metal is ~90 μm, the laser welded alloy retains ~98% of its YS. These differences indicate that the degree of reduction in YS after laser welding decreases when the grain size of the base metal increases.

The grain sizes of the columnar grains in the weld are larger than those of equiaxed grains in base metal, leading to a reduction in YS. When the grain size of base metal is small (e.g., ~5 μm and ~12 μm), coarse columnar grains in the weld result in a significant increase in grain size for the welded alloy, leading to a significant reduction in YS. Since grain sizes of columnar grains in the welds are similar, when the grain size of the base metal is initially

large (e.g., ~90 μm), the difference in grain sizes between weld and base metal is relatively small. Thus, the degree of reduction in YS decreases when the grain size of the base metal increases, as observed in Figure 8a.

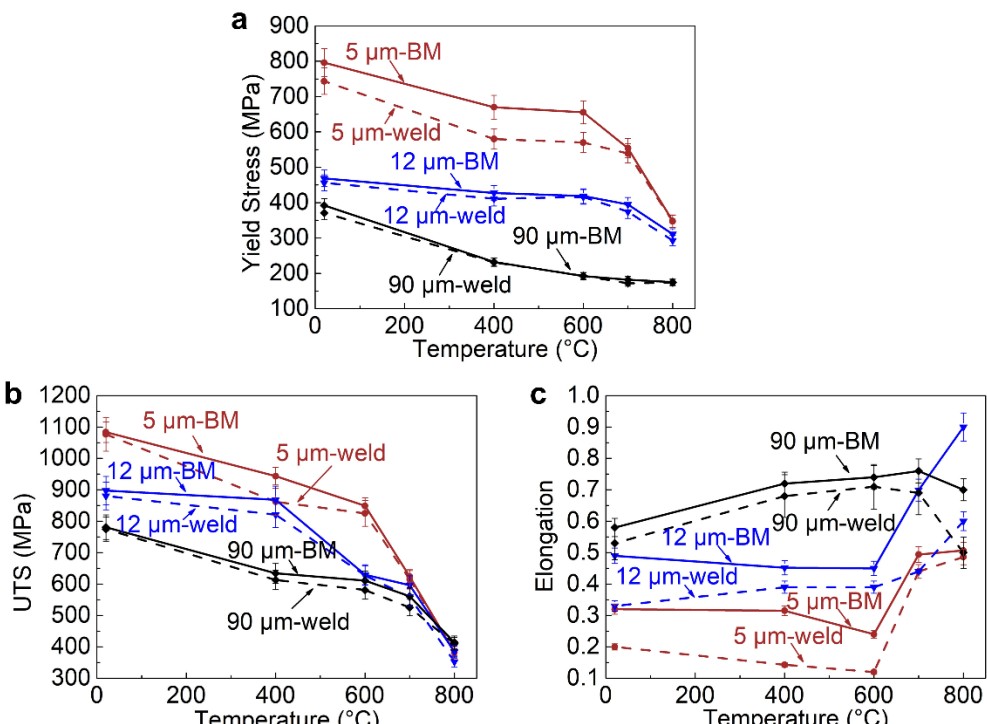

**Figure 8.** Average values of (**a**) YS, (**b**) UTS, and (**c**) EF of Hastelloy X base metals and welded alloys at various temperatures from 20 °C to 800 °C. The degrees of reduction in YS, UTS, and EF after welding all reduce when the grain size of the Hastelloy X base metal increases.

Similar to the reduction in YS, the degree of reduction in UTS of the welded Hastelloy X alloy also decreases when the grain size of base metal increases from ~5 μm to ~90 μm (Figure 8b). However, even if the grain size of base metal is small, e.g., ~5 μm, the welded alloy can maintain nearly all of the tensile strength of the base metal.

The average values of the EF (Figure 8c) shows that the EF of the Hastelloy X alloy also reduces after laser welding. It is found that the degree of reduction in EF is also affected by the grain size of base metal. As observed in Figure 8c, at RT, compared with base metal, the welded alloys maintain ~60%, ~68%, and ~90% of the EF of the base metal, when grain size of base metal is ~5 μm, ~12 μm, and ~90 μm, respectively.

Due to the structural difference between the weld and base metals, the microstructure uniformity of the alloy reduces after welding. As a result, cracks are easier to concentrate inside the welds during tensile testing, resulting in a decrease in EF. In this case, the degree of reduction in EF is dictated by the degree of reduction in microstructure uniformity, which is affected by the grain size of the base metal. When the grain size of the base metal is small (e.g., ~5 μm and ~12 μm), great structural differences between the weld and base metal lead to a significant reduction in the microstructure uniformity, resulting in a significant reduction in EF; when the grain size of the Hastelloy X base metal is initially large (e.g., ~90 μm), the microstructure after welding is relatively uniform, resulting in a slight reduction in the EF. Thus, the degree of reduction in the EF decreases when grain size of base metal increases, as observed in Figure 8c.

## 5. Summary and Conclusions

The microstructure and mechanical properties of a welded Hastelloy X superalloy are investigated systematically. The hot-rolled Hastelloy X alloy bars are homogenized,

cold-rolled, and then recrystallized to obtain equiaxed grain microstructures with average grain sizes of ~5 μm, ~12 μm, and ~90 μm. Laser welding processes are used to join sheets of the alloy, and then the alloy's weldability is determined through microstructural and mechanical property characterizations on the base metal and welds. Moreover, the effect of microstructure, especially grain size, on the weldability of the Hastelloy X alloy is investigated. The following conclusions can be drawn:

(1) After laser welding, there is no cracking in any welded alloys. The microstructures in the weld consist of columnar grains, which grow in the direction from the fusion line to the centerline.

(2) The sizes of the columnar grains in the weld are almost the same when the grain size of the Hastelloy X base metal increases from ~5 μm to ~90 μm. There is Mo-rich segregation in the fusion zone during welding, but this segregation is not a laves phase.

(3) The fractures of both base metals and welded alloys are ductile. The fractures tend to take place at the weld center due to the larger grain sizes of columnar grains at this area.

(4) Compared to the base metal, the YS, UTS, and EF of the Hastelloy X alloy all reduce after laser welding. When the grain size of the Hastelloy X base metal increases, the increase in grain size for the welded alloy becomes *small*, and therefore the degree of reductions in YS and UTS decreases; In the meantime, the decreases of EF also become small.

**Author Contributions:** Y.L.: formal analysis, investigation, writing—original draft, writing—review and editing. Q.D.: conceptualization, methodology, formal analysis, investigation, writing—original draft, writing—review and editing. X.W.: resources, writing—review and editing. Y.Z.: resources, writing—review and editing. Z.Z.: conceptualization, formal analysis, resources, writing—review and editing, supervision. H.B.: conceptualization, methodology, formal analysis, investigation, resources, writing—original draft, writing—review and editing, supervision. All authors have read and agreed to the published version of the manuscript.

**Funding:** This work is supported by Basic Science Center Program for Multiphase Media Evolution in Hypergravity of the National Natural Science Foundation of China (No. 51988101), the Key R & D Project of Zhejiang Province (No. 2020C01002), Natural Science Foundation of Zhejiang Province (No. LQ20E010008), National Science and Technology Major Project of China (J2019-III-0008-0051), the Innovation Fund of the Zhejiang Kechuang New Materials Research Institute (No. ZKN-20-P01, ZKN-20-Z01).

**Institutional Review Board Statement:** Not applicable.

**Informed Consent Statement:** Not applicable.

**Data Availability Statement:** The data presented in this study are available on request from the corresponding author.

**Conflicts of Interest:** The authors declare no conflict of interest.

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
