# Peer review of "The Microstructures and Mechanical Properties of a Welded Ni-Based Hastelloy X Superalloy"

_crystals, doi:10.3390/cryst12101336_

Round 1

Reviewer 1 Report

Microstructure and mechanical properties of a welded Hastelloy X superalloy are investigated and well organized results are presented. I recommend that this paper is suitable for publication in Crystals.

Please consider following one minor point before publication.

• Line 90: The strain rate seems to be 0.57/90=6x10^-2/s, not 10^-3/s.

Author Response

Microstructure and mechanical properties of a welded Hastelloy X superalloy are investigated and well-organized results are presented. I recommend that this paper is suitable for publication in Crystals.

Reply: Thanks very much for the reviewer’s positive comment.

Please consider following one minor point before publication.

Line 90: The strain rate seems to be 0.57/90=6x10^-2/s, not 10^-3/s.

Reply: Thank the reviewer for pointing out this mistake. Here, we made a careless error in the unit of crosshead displacement rate, which should be mm/min instead of mm/s. Therefore 0.57mm/min corresponding to an engineering strain rate of 10-3 s-1. We have corrected this in the revised manuscript.

Reviewer 2 Report

I could not understand why laser welded samples show different grain sizes because the grain sizes of the base metals are not different significantly (5 to 90-micron meter).  

Author Response

I could not understand why laser welded samples show different grain sizes because the grain sizes of the base metals are not different significantly (5 to 90-micron meter).

Reply: Thank the reviewer for pointing out this mistake. There is a fault in our original 90 micron welded microstructures. We have made corrections in the revised manuscript. The results show that the size of columnar grain in the weld does not change significantly when the grain size of the base metal changes. We apologize for the mistakes.

Reviewer 3 Report

Respected sir

The manuscript was reviewed carefully it may be considered for publication after making the following corrections

They conducted laser welding of Ni based alloys and results are acceptable. However, did they observe any laves formation in the fusion zone? Or not what is the reason?  that should be given

First point of conclusion to be re written

Author Response

They conducted laser welding of Ni based alloys and results are acceptable. However, did they observe any laves formation in the fusion zone? Or not what is the reason? that should be given.

Reply: Thanks for the reviewer’s positive comments. We have added a sentence in section 3.3 to point out that no laves phase is observed in the fusion zone.

First point of conclusion to be re written

Reply: Thanks very much for the reviewer’s comments. The first point of conclusion has been revised.